# Towards Efficient Image Compression Without Autoregressive Models

**Muhammad Salman Ali** [1], **Yeongwoong Kim**[1], **Maryam Qamar** [1],
**Sung-Chang Lim**[2], **Donghyun Kim**[2], **Chaoning Zhang**[1], **Sung-Ho Bae**[*1], **Hui Yong Kim**[*1]

[1] Kyung Hee University, Republic of Korea
[2] Electronics and Telecommunications Research Institute (ETRI), Republic of Korea
{salmanali, duddnd7575, maryamqamar}@khu.ac.kr,
{sclim, kimddng}@etri.re.kr,
chaoningzhang1990@gmail.com
{shbae, hykim.v}@khu.ac.kr

## Abstract

Recently, learned image compression (LIC) has garnered increasing interest with its rapidly improving performance surpassing conventional codecs. A key ingredient of LIC is a hyperprior-based entropy model, where the underlying joint probability of the latent image features is modeled as a product of Gaussian distributions from each latent element. Since latents from the actual images are not spatially independent, autoregressive (AR) context based entropy models were proposed to handle the discrepancy between the assumed distribution and the actual distribution. Though the AR-based models have proven effective, the computational complexity is significantly increased due to the inherent sequential nature of the algorithm. In this paper, we present a novel alternative to the AR-based approach that can provide a significantly better trade-off between performance and complexity. To minimize the discrepancy, we introduce a correlation loss that forces the latents to be spatially decorrelated and better fitted to the independent probability model. Our correlation loss is proved to act as a general plug-in for the hyperprior (HP) based learned image compression methods. The performance gain from our correlation loss is 'free' in terms of computation complexity for both inference time and decoding time. To our knowledge, our method gives the best trade-off between the complexity and performance: combined with the Checkerboard-CM, it attains **90%** and when combined with ChARM-CM, it attains **98%** of the AR-based BD-Rate gains yet is around **50 times** and **30 times faster** than AR-based methods respectively.

## 1 Introduction

In the digital world, visual media has proven to be a mixed blessing. On the one hand, it provides a mechanism to disseminate a plethora of data, information, and knowledge in a form easily consumable by humans Silk et al. [2021], while on the other, improved quality and increased content is very quickly pushing the boundaries of data storage, communication, and processing technologies Kim et al. [2021]. A software-driven solution to skew the balance of this state has been extensively presented in the form of lossy compression techniques Wallace [1991], Rabbani and Joshi [2002], Ballé et al. [2017]. In the case of images, these techniques aim to reduce the data size while maintaining its context. Traditional image codecs like JPEG Wallace [1991] and JPEG2000 Rabbani and Joshi [2002] use mathematically well-defined transformations such as discrete cosine transform (DCT) and discrete wavelet transform (DWT). These context-less transformations lead to reduced efficiency in terms of maintaining an optimal balance between the image quality and bitrate of the encoded image. In complex and varied images, this problem is especially pronounced, leading to the inclusion of various artifacts such as ringing and blocking Ballé et al. [2017]. In recent years,

---

[*]Corresponding Authors

37th Conference on Neural Information Processing Systems (NeurIPS 2023).

deep learning has significantly advanced several computer vision tasks Hu et al. [2022], especially image compression, by incorporating structure and learned behavior. In particular, learned image compression (LIC) has gained popularity as a field of research that has the potential to go beyond traditional compression methods by exploring the inherent nature of images Toderici et al. [2016], Lee et al. [2022], Johnston et al. [2018], Qian et al. [2022], Cheng et al. [2019].

State-of-the-art learned image compression techniques employ transform coding strategies for lossy image compression, which maps image pixels into a quantized latent space Cheng et al. [2020], Zhong et al. [2020]. This map is then losslessly compressed using entropy coding methods such as Huffman or arithmetic coding to create a bitstream for transmission over a channel Shannon [1948]. The entropy coding algorithm requires an entropy model, which is a prior probability model on the quantized latent representation. In this framework, deep neural networks are used for both transform coding and entropy modeling. By learning both modules end-to-end, the ability of deep neural networks is fully utilized to improve the compression performance Yang et al. [2023]. The bitstream size depends on the accuracy of the entropy model, making it essential to design an accurate model for efficient compression Kim et al. [2022]. The objective of the entropy model is to estimate a joint probability distribution over the elements of the quantized latent representation. A simplistic approach is to assume complete independence among the elements, but this approach results in subpar compression performance Ballé et al. [2017, 2018]. Thus, modeling the quantized latent representation interdependencies in learned image compression is a significant challenge Lee et al. [2019], Minnen et al. [2018], Qian et al. [2021], Theis et al. [2017].

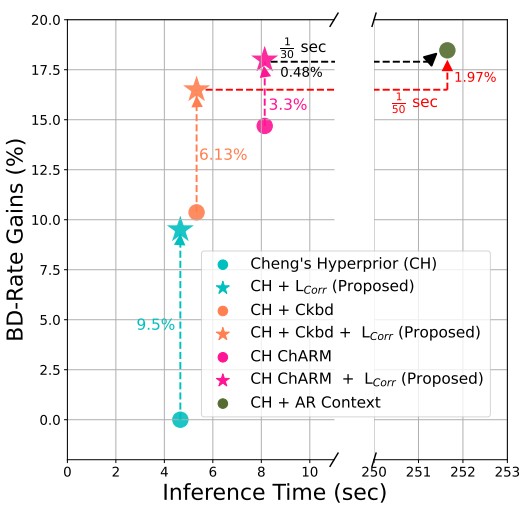

Figure 1: Performance-complexity tradeoff using various entropy models on the top of Cheng's architecture. Incorporating correlation loss to Cheng's hyperprior yields a BD rate gain of 9.5%, comparable to the gains achieved by the Checkerboard (CKBD) method. Applying correlation loss to the CKBD model results in a BD rate gain of 6.13% in comparison to the CKBD baseline, signifying approximately 90% of the gain seen in AR-based models. Notably, this improvement is achieved with a computational expense merely 1/50th that of the full AR-based model. Similarly, incorporating correlation loss into ChARM leads to a BD rate gain of 3.3% over the ChARM baseline, akin to 98% of the gain achieved by a full AR model, while utilizing only 1/30th of the computational resources.

A common approach is to use convolutional neural networks (CNNs) to extract additional features, known as "hyperprior," which captures the context from the quantized latent representation to improve the efficiency of the entropy model Ballé et al. [2018]. The underlying assumption of hyperprior-based entropy models is the probabilistic independence of the latent features, which contradicts the highly correlated nature of latent space, resulting in a discrepancy between the actual distribution and the assumed probability model Kim et al. [2022]. To minimize this discrepancy, Autoregressive context model (AR-CM) based LIC methods have been proposed Minnen et al. [2018], where the probability of each latent is updated based on the previously coded neighboring latents. While this increases the model's overall performance, it introduces sequential dependencies, making it roughly $50\times$ slower than simple hyperprior-based models He et al. [2021] and hinders its practical implementation of neural image compression in real-world scenarios

To mitigate the above issue, a line of works have attempted to discard AR method for improving efficiency. A notable advancement in this domain is the channel-wise autoregressive entropy model (ChARM) introduced by Minnen et al. Minnen and Singh [2020]. ChARM focuses on reducing the element-level serial processing within the context model presented in Minnen et al. [2018], and it demonstrates a reasonably good RD performance compared to a full autoregressive context model. Li et al. introduced CCN Li et al. [2020], which enables faster context calculation with moderate parallelizability. However, the effectiveness of CCN is still constrained by the image size, limiting its overall efficiency. He et al. He et al. [2021] proposed a parallelizable checkerboard context model

along with a two-pass decoding method, aiming to achieve a better trade-off between rate distortion (RD) performance and computational efficiency. The efficiency of the above methods is increased at a cost of performance drop compared to AR-based models.

To this end, we introduce a correlation loss to minimize the discrepancy without incurring any sequential dependency. The correlation loss forces latents to be decorrelated spatially, resulting in a better fit with the spatially independent probability model. Our findings indicate that decorrelating the latent aids in enhancing the general effectiveness of the hyperprior entropy model. Figure 1 illustrates the effectiveness of the proposed approach. Applying correlation loss to Cheng's hyperprior model yields a comparable performance improvement to that achieved by integrating efficient context modeling approaches. When our approach is integrated with the Checkerboard method, it leads to substantial BD rate improvements of approximately 90%, and when coupled with ChARM, it exhibits BD rate gains of about 98%, akin to the gains observed in a full AR-CM. Notably, our method achieves these remarkable performance gains while requiring only 1/50th and 1/30th of the inference time required by the AR model, respectively. Our key contributions can be summarized as follows:

- Our work takes the path of decorrelating latents that have never been tried in the field. The previous approaches primarily focus on CMs to enhance their performance, whereas, in our work, we address the fundamental need for employing computationally heavy CMs. With the introduction of our correlation loss, the correlation among spatially-neighbored elements in the latent space is decreased. This would minimize the discrepancy between the hyperprior entropy model's assumed probability distribution and the actual distribution of the latents, which eventually leads to an overall improvement of the rate-distortion (RD) performance.

- Since our method modifies only the loss function, i.e. adding a correlation loss to the original one, it does not require any additional memory or computational complexity beyond the base model. In contrast to the AR-based method which has severe sequential dependency among neighboring latent elements and thus could increase the inference time or decoding time up to $100\times$ the base model, our method does not create any sequential dependency and thus can maintain the inference or decoding time of the base model.

- Our proposed correlation loss acts as a plug-in for the existing learned image compression methods and can achieve BD rate gains of up to 9.5% by combining with the base hyperprior-based models. Remarkably, the gains are comparable to the performance improvements achieved by combining the hyperprior module with efficient context-based LIC models, like the SOTA efficiency-oriented Checkerboard's context model (CM). When combined with Checkerboard method, our proposed approach acheives 90% and when combined with ChARM our proposed approach acheives 98% of the AR-CM's BD-Rate gains while incurring only 1/50th and 1/30th of the inference time of the AR-CM. Our method achieves the best trade-off between the complexity and performance.

## 2 Related Work

The autoregressive context based entropy model, inspired by the concept of context used in traditional codecs, is used to predict the probability of unknown codes based on latents that have already been decoded Bellard [2015]. These models use hyper latent and context to predict both the location (mean value) and scale parameters (standard deviation) of the entropy model Minnen et al. [2018]. By combining elements of differentiable entropy modeling, hyper latent, and context models, it is possible to outperform BPG Bellard [2015] in terms of PSNR and MS-SSIM Minnen et al. [2018], He et al. [2021], Qian et al. [2022]. These context models have also been extended to become more powerful but also more computationally expensive Cheng et al. [2020]. In particular, AR context-based models, essential for achieving the state of the art performance, have low computational efficiency due to the sequential dependency of decoding the latents. As AR context models are sequentially dependent, they cannot be deployed on edge devices or take advantage of parallel hardware processing. As a result, models focusing on the real-world deployment of learned image compression often exclude AR context models due to their inefficiency and high inference costs He et al. [2021].

Parallel decoding approaches like Checkerboard He et al. [2021] and ChARM Minnen and Singh [2020] offer a better performance tradeoff over computation complexity. Checkerboard introduces a parallel decoding approach by encoding and decoding only half of the latent variables using a checkerboard-shaped context and hyperprior. The remaining latents, referred to as anchors, are coded solely based on the hyperprior. On the other hand, ChARM utilizes channel conditioning instead of

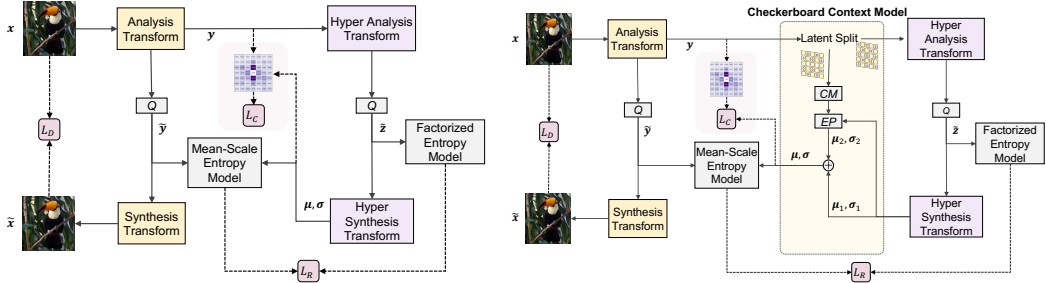

Figure 2: Operational diagram combining Hyperprior Ballé et al. [2018] with our proposed correlation loss

Figure 3: Operational diagram combing Checkerboard method He et al. [2021] with our proposed correlation loss

element-wise conditioning, along with latent residual prediction, to enhance the rate-distortion (RD) performance. The recent efficient SOTA methods are all built on top of the ChARM and checkerboard context model as an alternative to a fully AR-based context model Zhu et al. [2022], Qian et al. [2022], Kim et al. [2022].

Recent research in the field of entropy coding has primarily concentrated on improving the efficiency of context-based methods. These methods aim to exploit the interdependencies within the latent space more effectively, utilizing the inherent dependencies present Lei et al. [2022], Lee et al. [2019]. However, there has been a lack of attention towards enhancing the effectiveness of the hyperprior module, an integral component of prominent learned image compression (LIC) techniques. Notably, recent studies utilizing Transformer Vaswani et al. [2017] models have demonstrated that achieving a higher level of decorrelation in the latent space leads to improved rate-distortion (RD) performance Zhu et al. [2022]. This improvement can be attributed to the reduction in the discrepancy between the actual distribution of the latent space and the assumed probability model Kim et al. [2022]. Our work aims to address this research gap by introducing a novel approach that builds upon the existing findings. Specifically, we incorporate correlation loss during the training phase of our model, which promotes better decorrelation among the latent variables. By doing so, we enhance the performance of the hyperprior module, thereby improving the overall efficacy of LIC models.

## 3 Methodology

### 3.1 Formulation of Learned Image Compression Models

Image compression can be formulated through the transform coding approach by (as Figure 2 and 3)

$$
\begin{aligned}
y &= g_a(x; \phi) \\
\hat{y} &= Q(y) \\
\hat{x} &= g_s(\hat{y}; \theta),
\end{aligned}
\tag{1}
$$

where $x$ and $\hat{x}$ are input and reconstructed image, $y$ is a latent representation, and $\hat{y}$ is the quantized latent representation. $\phi$ and $\theta$ are the parameters of analysis and synthesis transforms $g_a$ and $g_s$, respectively, $Q$ represents the quantization. During training, the quantization is approximated using uniform noise. While for inference, a round function is applied for quantization to generate $\hat{y}$, and then entropy encoded to generate the bitstream. Given a probability model for the quantized representation, entropy encoding techniques such as arithmetic coding Rissanen and Jr. [1981] can losslessly compress the quantized codes. Arithmetic coders are optimal entropy coders which can be used to estimate the entropy of $\hat{y}$ during training.

A prior work Ballé et al. [2017] has employed a non-adaptive density model which was shared between the encoder and decoder also called factorized prior. It proposed a hyperprior in the follow-up work as a mechanism to introduce side information $z$ to capture the spatial dependencies in the latent space $y$ Ballé et al. [2018]. The side information predicted the mean and scale information for the latent space as:

$$
\begin{aligned}
z &= h_a(y; \phi_h) \\
\hat{z} &= Q(z) \\
p_{\hat{y}|\hat{z}}(\hat{y} \mid \hat{z}) &\leftarrow h_s(\hat{z}; \theta_h),
\end{aligned}
\tag{2}
$$

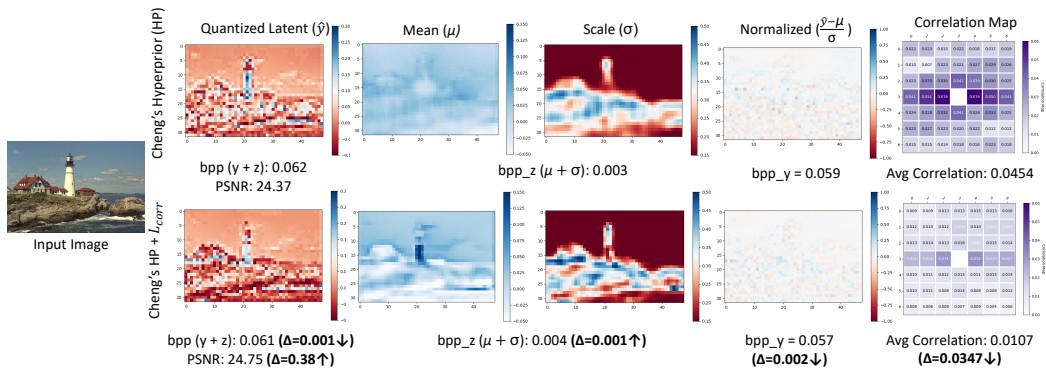

Figure 4: Visualization of quantized latent $\hat{y}$, hyperprior ($\mu$ and $\sigma$), normalized latent, and correlation map using Cheng's Hyperprior and Cheng's Hyperprior with correlation loss using an image from the Kodak dataset. The application of correlation loss enhances the details of the hyperprior and significantly reduces the correlation of latent variable $\hat{y}$, leading to improved compression efficiency and reconstruction quality.

where $h_a$ and $h_s$ denote the analysis and synthesis transform for the hyperprior auto-encoder. $p_{\hat{y}|\hat{z}}(\hat{y} \mid \hat{z})$ is the estimated probability distribution for the quantized latent space $\hat{y}$, given by:

$$p_{\hat{y}|\hat{z}}(\hat{y}_i \mid \hat{z}) = \left[ \mathcal{N}\left(\mu_i, \sigma_i^2\right) * \mathcal{U}\left(-\frac{1}{2}, \frac{1}{2}\right) \right](\hat{y}_i) \tag{3}$$

$$p_{\hat{y}|\hat{z}}(\hat{y} \mid \hat{z}) = \prod_i p_{\hat{y}|\hat{z}}(\hat{y}_i \mid \hat{z}), \tag{4}$$

where the mean $\mu_i$ and scale $\sigma_i$ parameter is the $i^{th}$ element of $h_s(\hat{z})$ for each spatial location $i$ in the latent space $\hat{y}$. With the premise that individual latent components are mutually independent, each spatial element in latent $\hat{y}$ is modeled as a Gaussian with its own mean and standard deviation. We may therefore estimate entropy using eq. 4. We modeled the probability of the hyper-latent $\hat{z}$ using a non-parametric fully factorized density model as:

$$p_{\hat{z}|\psi}(\hat{z} \mid \psi) = \prod_i \left( p_{z_i|\psi}(\psi) * \mathcal{U}\left(-\frac{1}{2}, \frac{1}{2}\right) \right)(\hat{z}_i), \tag{5}$$

where $z_i$ denotes the $i^{th}$ element of $z$, and $i$ specifies the position of each element, and the vectors $\psi(i)$ encapsulate the parameters of each univariate distribution $(p_{z_i|\psi}(\psi))$.

A context model can be added to boost the RD performance in the mean-scale hyperprior framework. Additionally, a parameter inference network ($g_{ep}$) is employed to estimate the mean and scale parameters $\Phi = (\mu, \sigma)$ by leveraging the outputs of the hyperprior module $h_s(\hat{z})$ and the context model $g_{cm}(\hat{y}_{<i})$:

$$\Phi_i = (\mu_i, \sigma_i) = g_{ep}(h_s(\hat{z}), g_{cm}(\hat{y}_{<i})), \tag{6}$$

where $\hat{y}_{<i}$ means the causal context of the nearby visible latents to the latent $\hat{y}_i$. The context model can be calculated as:

$$g_{cm}(x) = (M \odot W)x + b, \tag{7}$$

where $\odot$ refers to the Hadamard product, b is a bias term. The variable M represents a mask commonly used to establish a top-left reference, necessitating strict Z-ordered serial decoding.

The Checkerboard context model He et al. [2021] introduces a parallel decoding approach where only half of the latent variables are encoded/decoded using a checkerboard-shaped context and hyperprior as shown in the Figure 3. The coding of the remaining half of the latents, referred to as anchors, relies solely on the hyperprior. To implement this approach, the eq. 6 above is modified as:

$$\Phi_i = \begin{cases} g_{ep}(h_s(\hat{z}), 0)_i, & \hat{y}_i \in \hat{y}_{\text{anchor}} \\ g_{ep}(h_s(\hat{z}), g_{cm}(\hat{y}_{\text{anchor}} ; M \odot W))_i, & \text{otherwise} \end{cases} \tag{8}$$

The masked convolution, denoted as $g_{cm}$, is applied conditioned on a checkerboard-shaped mask $M$, as described in eq. 7. The input for $g_{cm}$, denoted as $\hat{y}_{anchor}$, corresponds to the set of anchors, and $i$ represents the index for the i-th element $\hat{y}_i$ in the latent variable $\hat{y}$. For methods that utilize

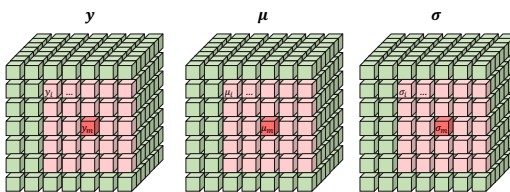

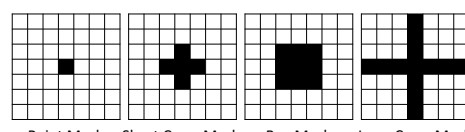

Point Mask   Short Cross Mask   Box Mask   Long Cross Mask

Figure 5: After normalizing the latent $\frac{y-\mu}{\sigma}$, the correlation will be calculated for all latents in the active $k \times k$ window, shown in pink, with the central point $m$.

Figure 6: Our proposed mask designs for the correlation loss. We have used Point Mask as a standard for all our experiments. Please refer to the supplementary material for the detailed ablation studies.

mean-scale Gaussian entropy models, the entropy parameter $\Phi$ is represented as $(\mu, \sigma)$, where $\mu$ and $\sigma$ correspond to the mean and standard deviation, respectively. In the presence of visible anchors, the context features of all non-anchors can be calculated simultaneously using masked convolution in parallel. The decoding of anchors also occurs in parallel. As a result, the calculation of entropy parameters in eq. 8 for decoding can be performed in two passes, which significantly improves efficiency compared to a serial context model.

LIC is a langrangian multiplier based rate-distortion optmization problem. The loss function is then defined as

$$
\begin{aligned}
\mathcal{L} =& \mathcal{R}(\hat{y}) + \mathcal{R}(\hat{z}) + \lambda \cdot \mathcal{D}(x, \hat{x}) \\
=& \mathbb{E}\left[-\log_2\left(p_{\hat{y}|\hat{z}}(\hat{y} \mid \hat{z})\right)\right] + \mathbb{E}\left[-\log_2\left(p_{\hat{z}|\psi}(\hat{z} \mid \psi)\right)\right] + \lambda \cdot \mathcal{D}(x, \hat{x}),
\end{aligned}
\tag{9}
$$

where $\lambda$ controls the rate-distortion tradeoff and $\mathcal{D}(x, \hat{x})$ is the distortion term usually using mean square error.

### 3.2  Our Proposed Correlation Loss

As shown in the eq. 4, the joint probability of $\hat{y}$ in the hyperprior-based entropy is calculated based on the assumption that the latent features are mutually independent. However, as shown in the upper portion of last column of Figure 4, the latent space is highly correlated, resulting in a discrepancy between the actual and the calculated probability distribution. To address this discrepancy, we introduce correlation loss to force the latent to be decorrelated spatially. We can therefore model the correlation loss by calculating the correlation in the latent space. We start by computing the correlation map as:

$$
Corr\_Map_{k \times k}[i] = \mathbb{E}_{x \sim p(x)}\left[\left(\frac{y_i - \mu_i}{\sigma_i}\right)\left(\frac{y_m - \mu_m}{\sigma_m}\right)\right], 0 \le i < k^2
\tag{10}
$$

where $k \times k$ is the window size of the correlation map, $\mu$ and $\sigma$ refer to the hyper-latent's corresponding $i^{th}$ and $m^{th}$ elements, $m$ refers to the central point of the window of size $k \times k$ as shown in Figure 5. We slide the window with the stride of 1 across the entire latent space to calculate the correlation of the central point $m$ with all the other points in the window of size $k \times k$. We then take the mean of all the individual correlation maps to get the resultant correlation map of the latent $y$ of size $k \times k$.

We then apply the $Mask$ on the correlation map. As shown in Figure 6, the point mask only masks the central location of latent space as it corresponds to self-correlation of 1 (when $i = m$). We propose several mask designs (discussed in supplementary material) to investigate the effect of the spatially relaxed decorrelation constraint. $Mask$ can then be applied as:

$$
Masked\_Map_{k \times k}[i] = Corr\_Map_{k \times k}[i] \odot Mask.
\tag{11}
$$

The correlation loss is formulated by applying the $L_2$ norm on the $Masked\_Map_{k \times k}$.

$$
L_{corr} = ||Masked\_Map_{k \times k}[i]||^2.
\tag{12}
$$

By incorporating the correlation loss $L_{corr}$, the RD loss can now be given as:

$$
RD_{loss} = E_{x \sim p_{(x)}}\left[-\log_2 p_{\hat{y}|\hat{z}}(\hat{y} \mid \hat{z}) - \log_2 p_{\hat{z}}(\hat{z})\right] + \lambda \cdot E_{x \sim p_{(x)}}[d(x, \hat{x})] + \alpha \cdot [L_{corr}],
\tag{13}
$$

where $\alpha$ is the scaling factor for the correlation loss.

Given that the entropy estimate presumes independent latent elements, our proposed loss function increases compression efficiency by reducing the correlation between latent features, hence reducing the discrepancy between the assumed probability distribution of the entropy model and the actual latent probability distribution.

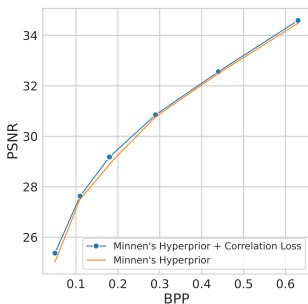 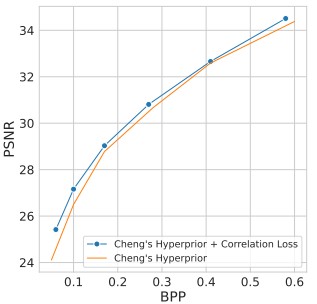 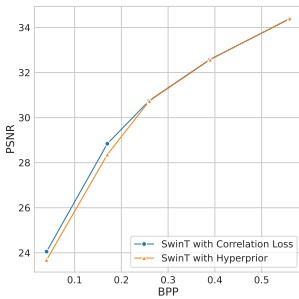

(a) Minnen's mean scale hyperprior  (b) Cheng's mean scale hyperprior  (c) SwinT mean scale hyperprior

Figure 7: RD rate comparison of (a) Minnen's (b) Cheng's and (c) SwinT mean scale hyperprior models with the inclusion of correlation loss.

# 4  Experiments and Results

## 4.1  Experimental Setup

We train all our models on the Vimeo-90k dataset Xue et al. [2019], with the images cropped to the resolution of $256 \times 256$ for training. The models were optimized using the Adam optimizer Kingma and Ba [2015] with a batch size of 16 and trained for 1.5 million iterations with a learning rate of $1 \times 10^{-4}$ for the first million iterations and then halved every 50,000 iterations till 1.25 million iterations.

RD loss given in eq. 9 is used to train the baseline models, whereas the modified loss in eq. 13 is used to train models with correlation loss. The rate-distortion tradeoff is guided by $\lambda$, whose value is contained in the set [0.0009, 0.0018, 0.0035, 0.0067, 0.0130, 0.0250].

We tested our model on a commonly used Kodak lossless images dataset Kodak [1993], with 24 uncompressed images of $768 \times 512$ or $512 \times 768$ resolution. Bits per pixel (bpp) provide the rate, while PSNR indicates the quality of the reconstructed image when evaluating rate-distortion performance. In order to demonstrate the coding efficiency, RD curves are drawn.

## 4.2  Rate-Distortion Performance

We employ Minnen's Minnen et al. [2018], Cheng's Cheng et al. [2020], and SwinT Liu et al. [2021] based mean scale hyperprior models as well as Cheng's with Checkerboard He et al. [2021] for our experiments. We perform all our experiments on the Pytorch framework Paszke et al. [2017] and use the CompressAI library Bégaint et al. [2020]. Minnen's and Cheng's models were trained using an NVIDIA 2080Ti, whereas the SwinT model was trained on an NVIDIA 3070Ti due to the transformers' high memory requirement. Note that including correlation loss has a negligible effect on the hardware requirement of these base models.

**Minnens' Hyperior:** Figure 7a depicts the RD curve for Minnen's model Minnen et al. [2018] with and without correlation loss; for Minnen's model, correlation loss achieves BD rate gains of 3.20%.

**Cheng's Hyperprior:** Figure 7b illustrates the RD curve for Cheng's model Cheng et al. [2020] with and without correlation loss. Cheng's model yields a BD rate gain of 9.5% with correlation loss. Cheng's model consists of attention modules along with CNN layers that significantly increase the overall efficiency of the correlation loss. Figure 9 depicts the spatial correlation of latent $y$ averaged across all latent channels and compares the baseline Cheng's model with the one trained on correlation loss. Figure 9 demonstrates that the model trained with our proposed loss function exhibits significantly less correlation than the baseline model. This observation holds for the models trained with other $\lambda$ values. The qualitative comparison in Figure 8 between Cheng's model with and without correlation loss demonstrates the superior performance achieved by incorporating correlation loss. It implies that models trained on correlation loss incur less redundancy across different spatial latent locations, resulting in a better rate-distortion trade-off overall.

**Transformer Hyperprior:** We also compared the effectiveness of correlation loss on the Transformer-based LIC model. We implemented our SwinT with a hyperprior model inspired by Zhu's work and employed a similar architectural setting Zhu et al. [2022]. Transformers tend to have a lesser correlation in their latent space than the convolutional-based LIC models Zhu et al. [2022]. The

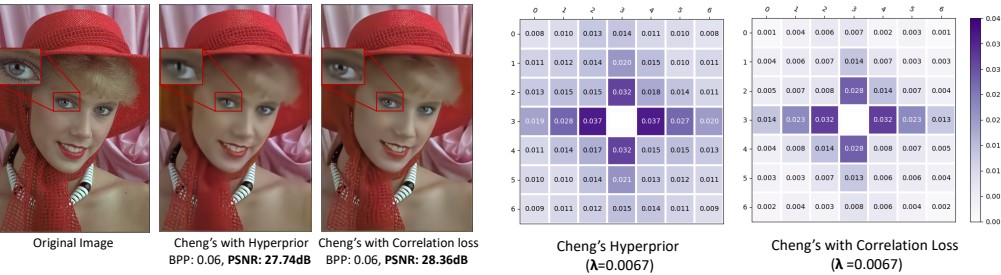

Figure 8: Comparison of reconstruction quality of the original image from Kodak dataset using Cheng's hyperprior and Cheng's hyperprior with correlation loss.

Figure 9: Spatial correlation map comparison of Cheng's hyperprior (left) with Cheng's hyperprior with correlation loss (right)

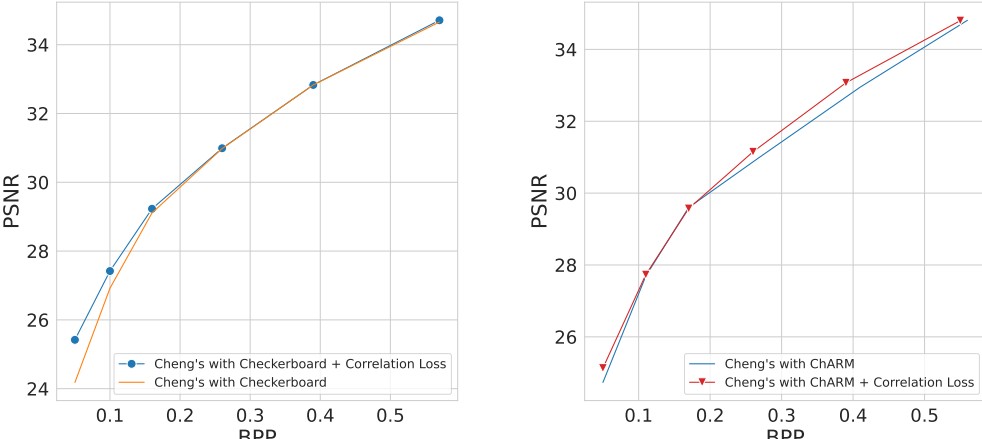

Figure 10: RD rate comparison of Cheng's + Checkerboard with the inclusion of correlation loss.

Figure 11: RD rate comparison of Cheng's + ChARM with the inclusion of correlation loss.

Figure 7c illustrates the performance of the SwinT model with and without correlation loss. We found that with the correlation loss, the SwinT model achieves a BD-rate gain of about 4.8%. The effectiveness of correlation loss for the SwinT model further indicates the efficacy of the correlation loss, which in turn improves the mean scale based entropy model.

**Cheng's with Checkerboard Context Model:** Figure 10 illustrates the RD curve for Cheng's model with Checkerboard He et al. [2021] with and without correlation loss. Cheng's with Checkerboard yields a BD rate gain of 6.13% with correlation loss. Checkerboard uses a parallel decoding approach by encoding and decoding only half of the latent variables using a checkerboard-shaped context. The remaining latents, referred to as anchors, are coded solely based on the hyperprior. Since half of the latents are encoded by the hyperprior module. Incorporating correlation loss into the Checkerboard context model significantly improves the performance of the hyperprior module, resulting in an overall enhancement of the model's performance.

**Cheng's with ChARM:** Figure 11 illustrates the RD curve for Cheng's model with ChARM Minnen and Singh [2020], both with and without the integration of correlation loss. Notably, when correlation loss is incorporated into Cheng's with ChARM, there is a substantial BD rate gain of 3.3%. ChARM effectively employs channel context and latent residual prediction for the encoding and decoding of the latents. The combination of correlation loss with ChARM leads to a significant enhancement in its overall performance, marking a performance difference of approximately 0.5% when compared to a full AR model.

## 4.3 Complexity Tradeoff

The primary advantage of the correlation loss lies in its capacity to minimize the discrepancy between the actual and presumed probability distribution within the entropy model. Figure 1 illustrates that for

Table 1: Average encoding and decoding time on the Kodak dataset for different entropy models applied to Cheng's Hyperprior. BD-Rate gains are evaluated using Cheng's Hyperprior as the baseline.

| Architecture | BD Rate Gains (%) | Inference Time (sec) |
|---|---|---|
| Cheng's Hyperprior (CH) | — | 4.66 |
| CH + Correlation Loss (Proposed) | **9.5** | 4.66 |
| CH + Checkerboard | 10.37 | 5.33 |
| CH + Checkerboard + Correlation Loss (Proposed) | **16.50** | 5.33 |
| CH + ChARM | 14.69 | 8.14 |
| CH + ChARM + Correlation Loss (Proposed) | **17.99** | 8.14 |
| CH + AR Context | 18.47 | 251.65 |

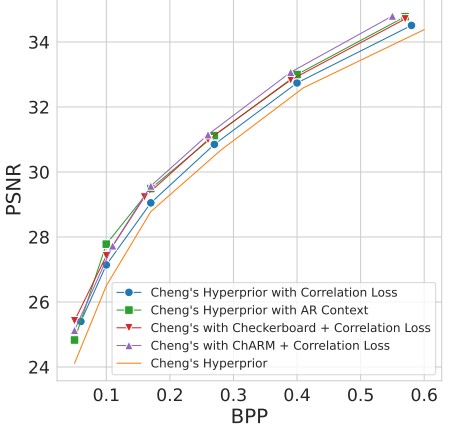

Figure 12: Comparison between Cheng's AR context, Cheng's hyperprior, Cheng's hyperprior with correlation loss, Cheng's Checkerboard with correlation loss, and Cheng's ChARM with correlation loss

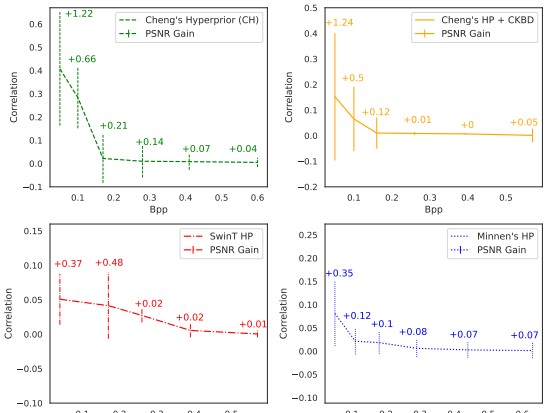

Figure 13: Visual representation depicting the relationship between bits per pixel (bpp) and correlation, and its influence on peak signal-to-noise ratio (PSNR) gains. For all the four models, both the average correlation and the gains decreases as the bpp value increases.

a given encoder and decoder architecture, which is Cheng's model in this case, the choice of entropy model determines the performance of the resulting model. Note that Cheng's hyperprior signifies the lower bound with the given encoder-decoder architecture, whereas Cheng's AR defines the upper limit with the same encoder-decoder architecture. Using Cheng's hyperprior model as the baseline, we compared the effectiveness of including an AR module vs. the correlation loss. The Figure 12 shows the RD curve comparison, while the Table 1 compares the inference time for processing the entire Kodak test dataset. Incorporating correlation loss into Cheng's hyperprior model results in a performance improvement roughly half that of the AR-context-based model, all while reducing inference costs to just 1/55th of the original. Similarly, when utilizing Cheng's Checkerboard with correlation loss, a substantial performance enhancement of approximately 90% is achieved compared to the AR-context-based model, with an inference cost as low as 1/50th of the latter. Additionally, Cheng's ChARM with correlation loss attains an impressive performance gain of about 98% when compared to the AR-context-based model, while incurring an inference cost of merely 1/30th of the AR-context-based model as shown in Figure 1.

It should be noted that when correlation loss is applied to Cheng's hyperprior, significant improvements can be achieved due to the large performance gap between Cheng's hyperprior baseline and the upper limit. On the other hand, since the gaps are diminished in models like Cheng's with Checkerboard and Cheng's with ChARM, the potential for improvement by applying correlation loss is also limited when compared to Cheng's hyperprior baseline. In summary, the introduction of correlation loss to these models would result in a closer approximation to the full AR performance, albeit with gains that are comparatively smaller.

# 5 Analysis

## 5.1 Visual Illustration of Effectiveness of the Correlation Loss

Figure 4 visualizes some of the internal workings of our models. We compare Cheng's hyperprior model with and without correlation loss. By observing the first column of Figure 4, it is clear that our correlation loss makes the latent feature preserve more detailed texture information yielding higher PSNR compared with the base model. The mean $\mu$ can also extract more distinctive and meaningful information from the latent space. Since the hyperpriors (mean $\mu$ and scale $\sigma$) contain more information, they require slightly more bits than the base model. Nevertheless, the rise in hyperprior rate is adequately offset by the relatively greater decrease in latent bitrate, as demonstrated in the final column of Figure 4, where the normalized latents exhibit improved decorrelation compared to the base model. Overall, our proposed method surpasses the base model by enhancing PSNR through more decorrelated latents while either maintaining or reducing bpp with better decorrelated latents.

## 5.2 Effect of Correlation score on the Model's performance

Our proposed loss function consistently demonstrates significant BD-rate gains at lower bit rates while maintaining comparable performance at high bit rates. This behavior is primarily attributed to learned image compression (LIC) models exhibiting higher correlation among latent variables at lower bpp values compared to higher bpp values as reported by Zhu et al. [2022] and also evident from the correlation maps in Figure 6 of the supplementary material. We conducted a comprehensive analysis and presented the findings in Figure 13. Figure 13 showcases the relationship between PSNR gains, bpp, and correlation for different models, including Cheng's hyperprior , Cheng's with Checkerboard, SwinT hyperprior, and Minnen's hyperprior. From the graphs in the Figure 13, a clear pattern can be observed: as the bpp decreases, the correlation of the latents increases, resulting in a higher gain in PSNR. However, as we move towards higher bpps, the correlation becomes notably reduced, resulting in decreased PSNR gain. This trend explains the reason why the efficacy of the correlation loss is larger in low bpp range and diminish at high bpp range.

# 6 Ablation Studies

Comprehensive ablation studies regarding various mask types, mask sizes, and $\alpha$ values are presented in the supplementary material.

# 7 Conclusion

The hyperprior-based entropy models assume probabilistic independence, which leads to a discrepancy between the actual and the assumed probability distribution. We proposed correlation loss, which reduces the correlation among spatially neighbored elements in the latent space resulting in a better fit with the spatially independent probability model. Our proposed loss function does not require any model structure or capacity changes and acts as a plug-in method for the existing neural compression methods. The proposed approach, without modifying the entropy models and increasing the inference time, significantly improves the RD performance with BD rate gains up to 17.99%.

## Acknowledgments and Disclosure of Funding

Authors are thankful to Muhammad Awais for his constructive and helpful feedback. Authors are also thankful to Taegoo Kang for managing and providing the required compute resources.
This work was supported in part by the Institute of Information and Communications Technology Planning and Evaluation (IITP) grant funded by the Korea Government [Ministry of Science and Information Communication Technology (MSIT)] (Development of Audio/Video Coding and Light Field Media Fundamental Technologies for Ultra Realistic Tera-media) under Grant 2017-0-00072; in part by the IITP grant funded by the Korea Government (MSIT) (Artificial Intelligence Convergence Innovation Human Resources Development, Kyung Hee University) under Grant RS-2022-00155911; in part by the IITP grant funded by the Korea Government (MSIT) (Artificial Intelligence Innovation Hub) under Grant 2021-0-02068.

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
