# Supplementary Material: Towards Efficient Image Compression Without Autoregressive Models

**Muhammad Salman Ali** [1], **Yeongwoong Kim**[1], **Maryam Qamar** [1],
**Sung-Chang Lim**[2], **Donghyun Kim**[2], **Chaoning Zhang**[1], **Sung-Ho Bae**[*1], **Hui Yong Kim**[*1]

[1] Kyung Hee University, Republic of Korea
[2] Electronics and Telecommunications Research Institute (ETRI), Republic of Korea

{salmanali, duddnd7575, maryamqamar}@khu.ac.kr,
{sclim, kimddng}@etri.re.kr,
chaoningzhang1990@gmail.com
{shbae, hykim.v}@khu.ac.kr

## A   Ablation Studies

**Effect of Alpha Value:** In the modified RD loss (eq.13 from the manuscript), $\alpha$ is used as a governing parameter to regulate the effect of correlation loss. The effect of regulating parameters is vital as its value can affect the effectiveness of the correlation loss. To determine the optimal values for $\alpha$ , we performed experiments using the point mask for different $\alpha$ values, specifically $1 \times 10^{-7}$, $1 \times 10^{-10}$, and $1 \times 10^{-13}$. The Figure 1 illustrates our experimental results. As shown in Figure 1, for lower BPPs, all the $\alpha$ values outperform Cheng's baseline. However, the $1 \times 10^{-13}$ $\alpha$ value performs slightly better than the other $\alpha$ values, with a BD rate gain of around 9.3%. The Figure 1 also indicates, specifically, at lower bpps, the overall performance difference between various alpha levels is not that substantial.

**Comparison of Different Mask Design:**  We designed different masks for $Mask$ as defined  in eq.11 based on the varying correlation patterns we observed in the prior research Zhu et al. [2022]. We use four mask types for our experiments: point mask, short-cross mask, long-cross mask, and border mask. Figure 6 (from the manuscript) illustrates our different mask types. To determine the feasibility and efficacy of various mask types, we conduct experiments on Cheng's model for various mask types and compare the results to the baseline. The masks used in the experiments had sizes of $3 \times 3$ for short-cross and point masks and $5 \times 5$ for long-cross and border masks. A $3 \times 3$ long-cross mask would be similar to a $3 \times 3$ short-cross mask, and a $3 \times 3$ border mask would mask the entire correlation map. The comparison of different mask designs with the baseline is given in the Figure 2. Regardless of the mask type, our proposed correlation loss outperforms the baseline by a significant margin, particularly at the lower bit rates where the correlation is highest Zhu et al. [2022].

**Changing the Window Size** ($k \times k$)**:**  Similar to the $\alpha$ value, the window size $k \times k$ of the correlation map can considerably affect the overall performance of the correlation loss. We conduct experiments with point mask, short-cross mask, long-cross mask, and border mask to analyze the influence of window size on correlation loss. Three different window sizes, specifically $3 \times 3$, $5 \times 5$, and $7 \times 7$, are used for this analysis. The results shown in Figure 3 indicate that the window size has little effect on the performance of the point mask, while the performance of the short-cross mask is similar for $3 \times 3$ and $5 \times 5$ windows but decreases at higher bit rates for a $7 \times 7$ window. The performance of the long-cross and border masks remains consistent for both $5 \times 5$ and $7 \times 7$ windows. These findings suggest that the window size does not significantly impact the performance of the correlation loss, but a $3 \times 3$ window size is preferred for its lower computational complexity during training.

---

[*]Corresponding Authors

37th Conference on Neural Information Processing Systems (NeurIPS 2023).

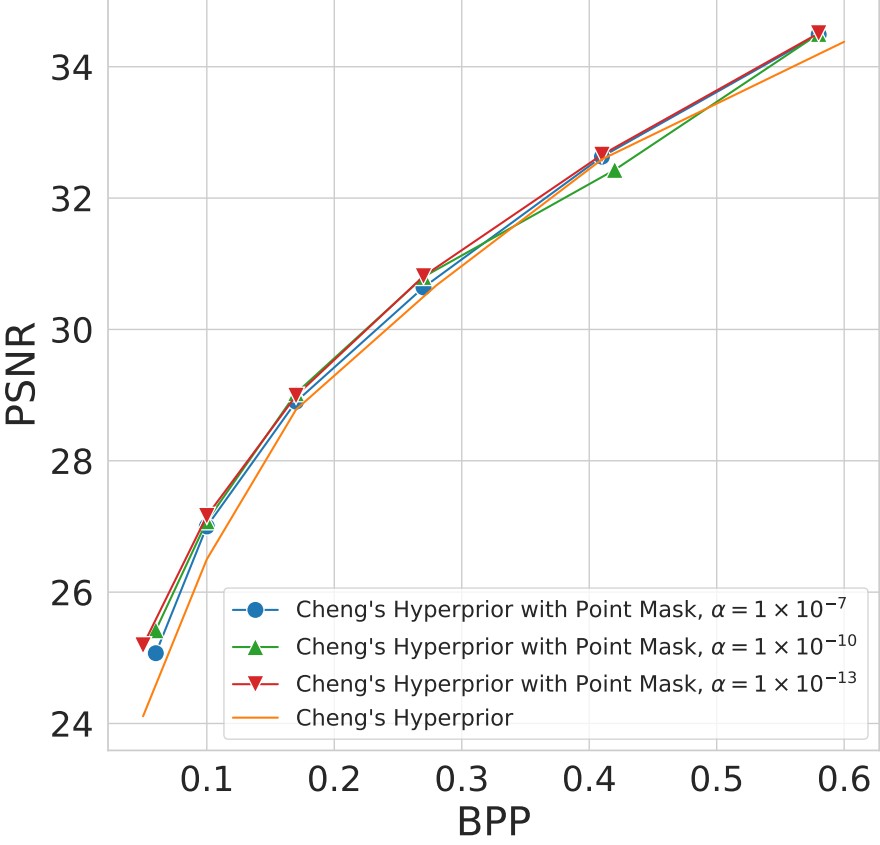

Figure 1: Impact of different $\alpha$ values on the correlation loss compared to the Cheng's baseline

## B  Additional Experimental Results and Visualizations

### B.1  Additional Quantitative Analysis

#### B.1.1  Minnen's and SwinT extended results

The plots in Figures 4 and 5 show the same results as Figure 7a and 7c from the main manuscript, while encompassing a wider range of compression methods and an extended bit range. Figure 4 provides a comparative evaluation of Minnen's hyperprior with correlation loss in contrast to Balle's scale hyperprior, Balle's factorized Prior, and Minnen's hyperprior with AR context model. Meanwhile, Figure 5 presents a comparison between SwinT hyperprior and SwinT hyperprior with correlation loss, specifically considering an extended bit range.

#### B.1.2  Comparison with ELIC

We conducted experiments on Efficient Learned Image Compression (ELIC) He et al. [2022](using the unofficial implementation available on Github[2]), investigating its performance both with and without the integration of the correlation loss. A comparison between ELIC and Cheng's hyperprior revealed a significant BD rate gain of 28.85% for ELIC. Intriguingly, when the correlation loss was introduced to ELIC, the BD rate gain was further elevated to approximately 30.36%. It is worth highlighting that the complete inference process for the ELIC on Kodak dataset took approximately 17.77 seconds which is about 1/15th of the total inference time of full AR. These findings strongly underscore the effectiveness of our proposed approach, showcasing its potential to achieve enhanced image compression performance. Table 1 shows the performance of ELIC with the inclusion of correlation loss.

---

[2]https://github.com/VincentChandelier/ELiC-ReImplemetation

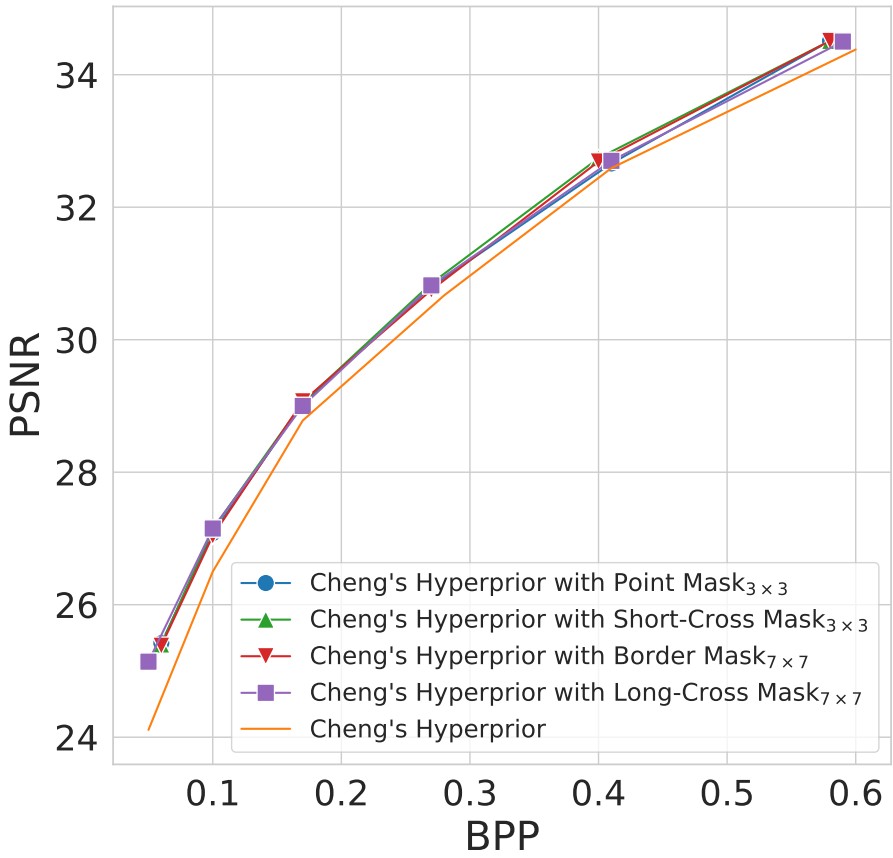

Figure 2: Comparison of different mask types with the Cheng's baseline

## B.2 Additional Correlation Maps Visualizations

The Figure 6 presents the spatial correlation maps for Cheng's hyperprior (HP), Cheng's HP with correlation loss, Cheng's HP with Checkerboard (CKBD) and Cheng's HP with CKBD and correlation loss on the Kodak dataset. The Figure 6 illustrates that for all $\lambda$ values, the model with correlation loss results in a considerably lower correlation when compared to the baseline models. Additionally, it is observed that at smaller $\lambda$ values, the correlation is at its highest, while with our approach, the correlation is significantly reduced. This leads to an improvement in overall performance, as seen by an increase in PSNR and a decrease in bpp.

## B.3 Visual illustration of the effectiveness of the correlation loss through visualizations of the latents and hyperpriors

Figures 1, 2, 3 demonstrate that our proposed method tends to improve the PSNR of the base model without increasing bpp. This tendency can be explained by visualizing the latents in Figure 7. By observing the first column, it is clear that our correlation loss makes the latent feature preserve more detailed texture information yielding higher PSNR compared with the base model. The mean $\mu$ can also extract more distinctive and meaningful information from the latent space. Since the hyperpriors (mean $\mu$ and scale $\sigma$) contain more information, they require slightly more bits than the base model. Nevertheless, the rise in hyperprior rate is adequately offset by the relatively greater decrease in latent bitrate, as demonstrated in the final column of Figure 7, where the normalized latents exhibit improved decorrelation compared to the base model. Overall, our proposed method surpasses the base model by enhancing PSNR through more decorrelated latents while either maintaining or reducing bpp with better decorrelated latents.

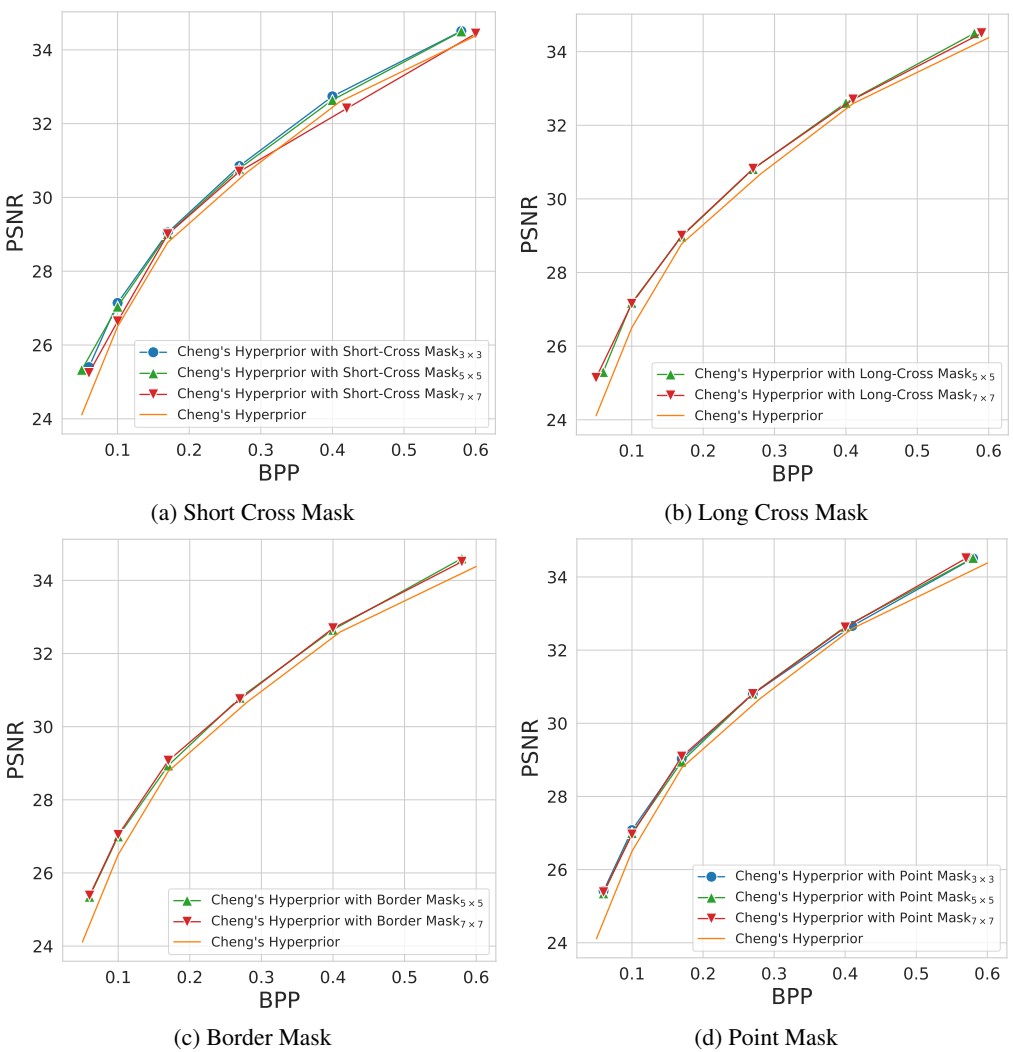

Figure 3: Comparison of correlation map window sizes using different masks on Cheng's hyperprior

## B.4 Additional Qualitative Comparison Results

Figure 8 presents a qualitative comparison between the SwinT hyperprior and the SwinT hyperprior with correlation loss. SwinT hyperprior exhibits artifacts that manifest as a textured pattern. On the other hand, SwinT with correlation loss does not have such artifacts, and it captures better features throughout all the reconstructed images. The Figure 9 provides a qualitative comparison of Cheng's HP, Cheng's HP with correlation loss, Cheng's HP with CKBD, and Cheng's HP with CKBD and correlation loss. The results clearly demonstrate that the inclusion of correlation loss improves the PSNR and reduces the bpp rate when compared to the baseline models.

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

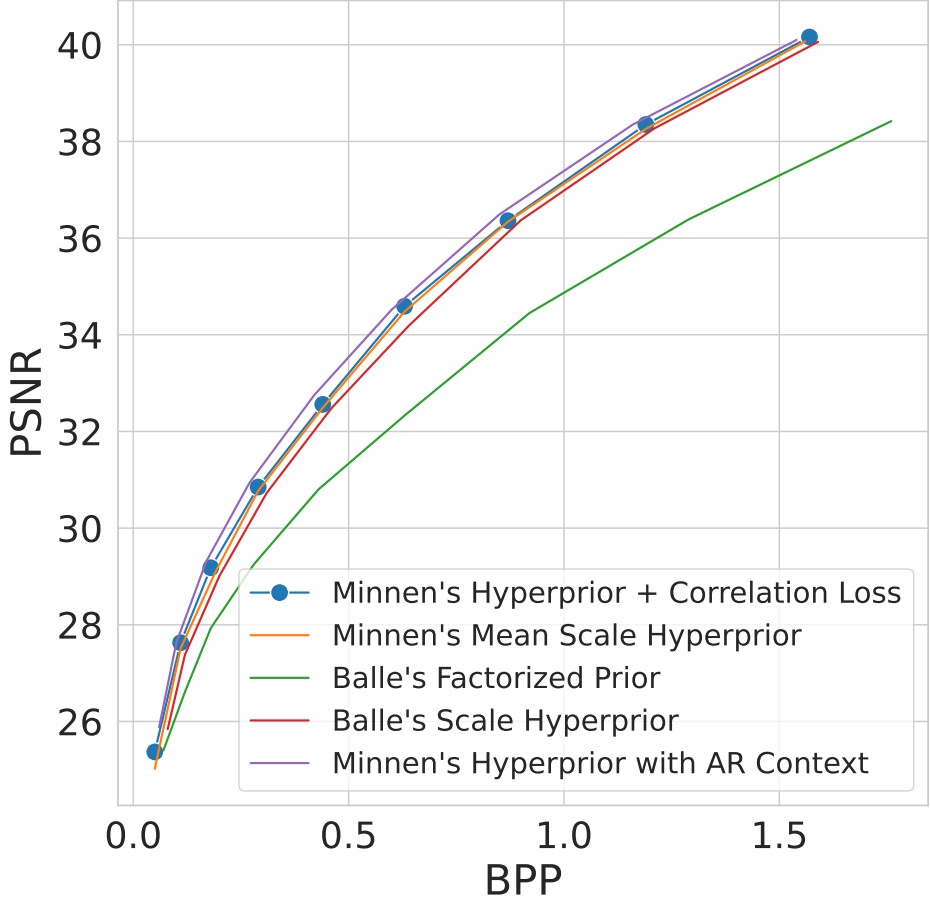

Figure 4: RD curve comparison between Minnen's AR context, Minnen's hyperprior, Minnen's hyperprior with correlation loss, Balle's scale hyperprior and Balle's factorized prior

| ELIC | | ELIC + $L_{corr}$ | |
|---|---|---|---|
| BPP | PSNR | BPP | PSNR |
| 0.06 | 27.72 | 0.06 | 27.77 |
| 0.11 | 28.95 | 0.10 | 28.94 |
| 0.18 | 30.36 | 0.18 | 30.38 |
| 0.28 | 31.77 | 0.28 | 31.78 |
| 0.43 | 33.73 | 0.43 | 33.74 |
| 0.61 | 35.34 | 0.61 | 35.40 |

Table 1: Performance Comparison of ELIC with and without correlation loss

OpenReview.net, 2022. URL https://openreview.net/forum?id=IDwN6xjHnK8.

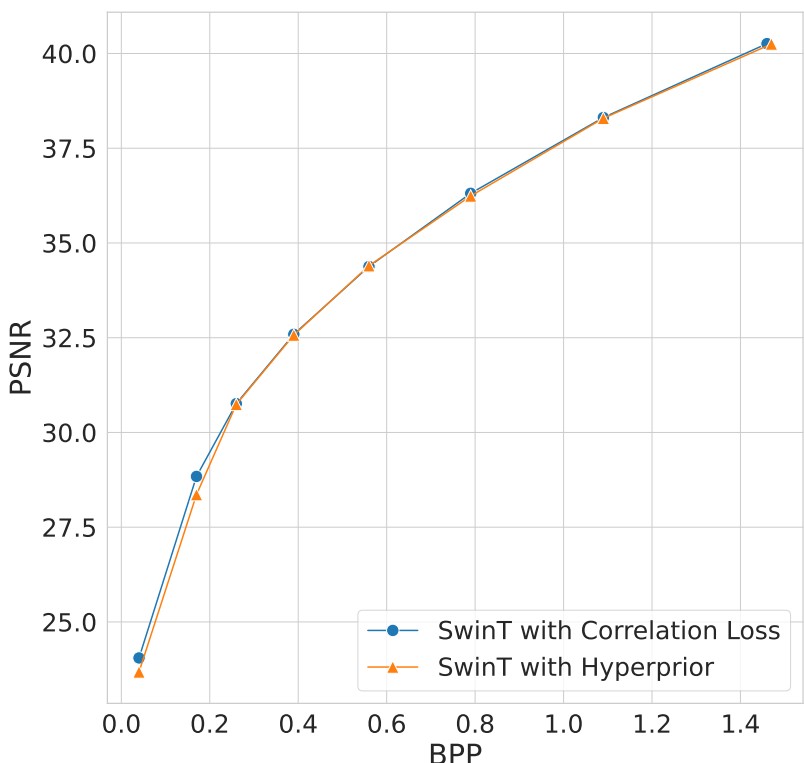

Figure 5: RD curve comparison of SwinT hyperprior with the inclusion of correlation loss

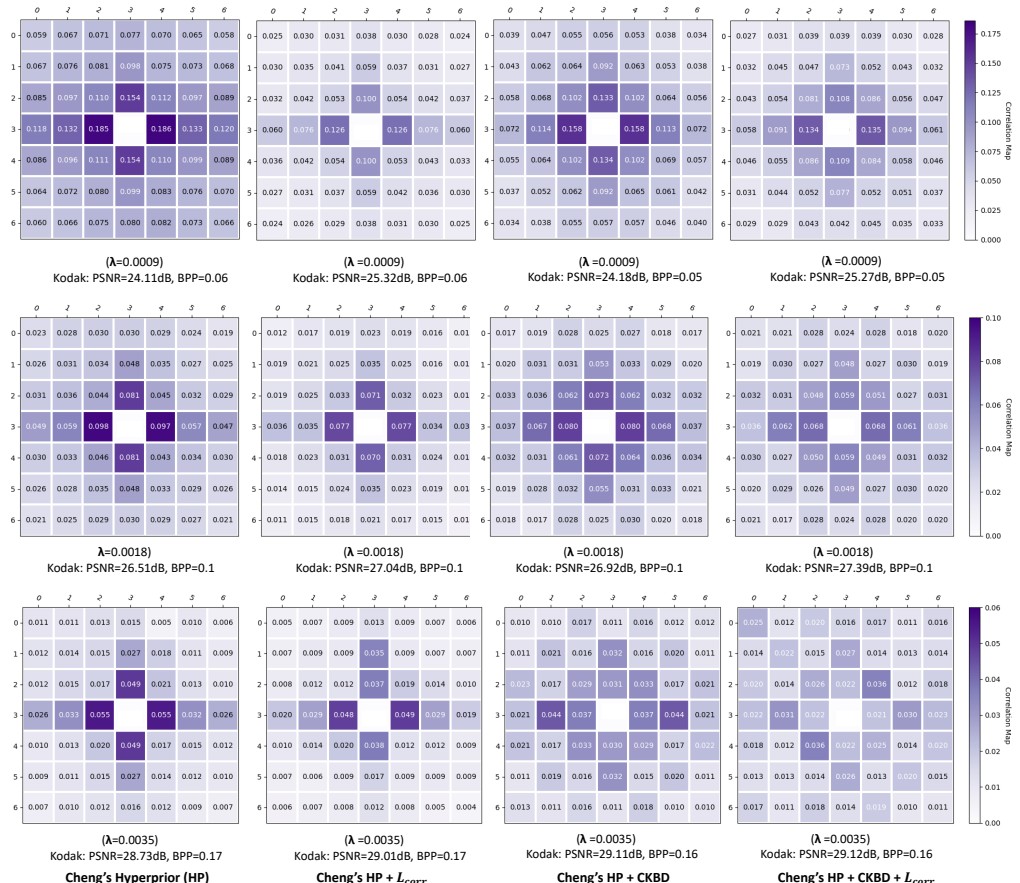

Figure 6: Spatial correlation of normalized latents averaged across all latent elements of all images on the Kodak dataset. The value at index (i, j) represents the normalized cross-correlation between latents at spatial positions (w, h) and (w + i, h + j). Each row in the figure presents a pair of models trained with the same $\lambda$ value, where the $\lambda$ values are 0.0009, 0.0018, and 0.0035. It is observed that the inclusion of correlation loss consistently results in decreased correlations when compared to the baseline models. Furthermore, the correlation increases for smaller $\lambda$ values (lower rate situations). However, in our approach, the correlation increases at a much slower rate when compared to the baseline model.

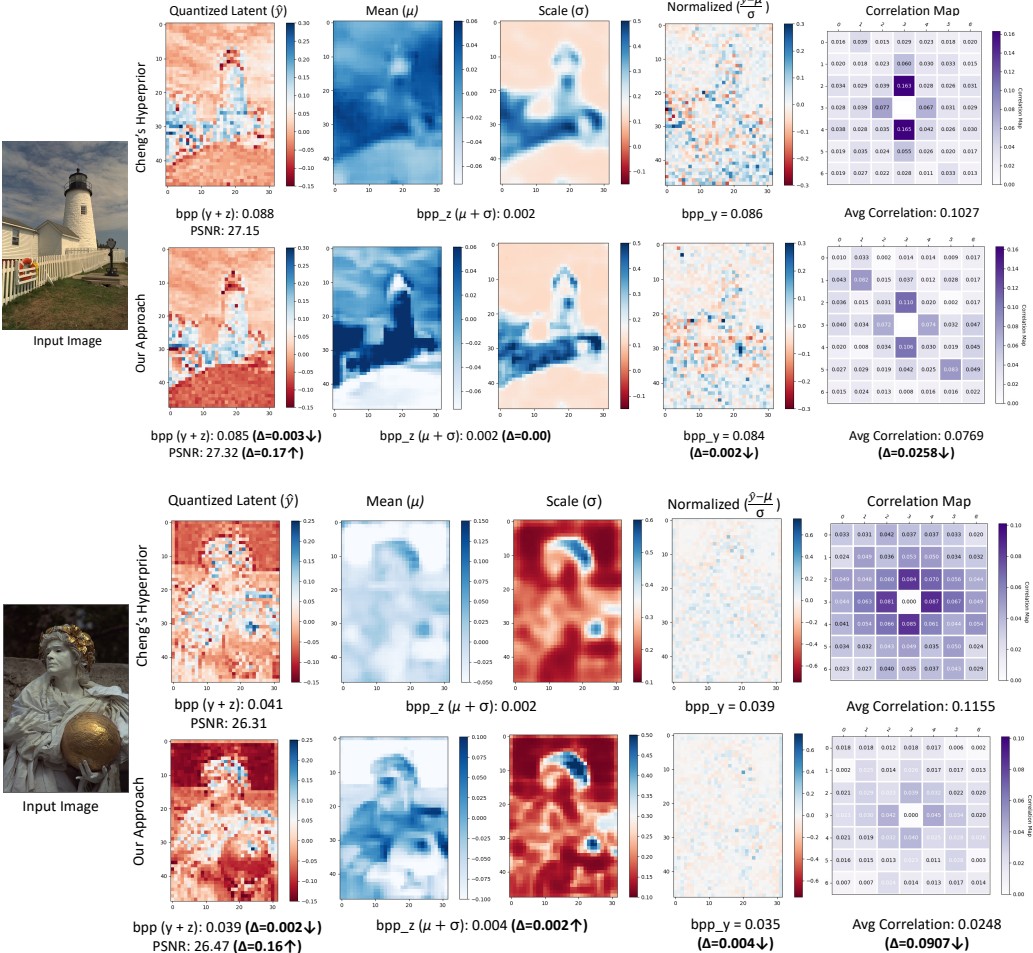

Figure 7: Visualization of quantized latent $\hat{y}$, hyperprior ($\mu$ and $\sigma$), normalized latent, and correlation map using Cheng's Hyperprior and Cheng's Hyperprior with correlation loss using images from the Kodak dataset. The application of correlation loss enhances the details of the hyperprior and significantly reduces the correlation of latent variable $\hat{y}$, leading to improved compression efficiency and reconstruction quality.

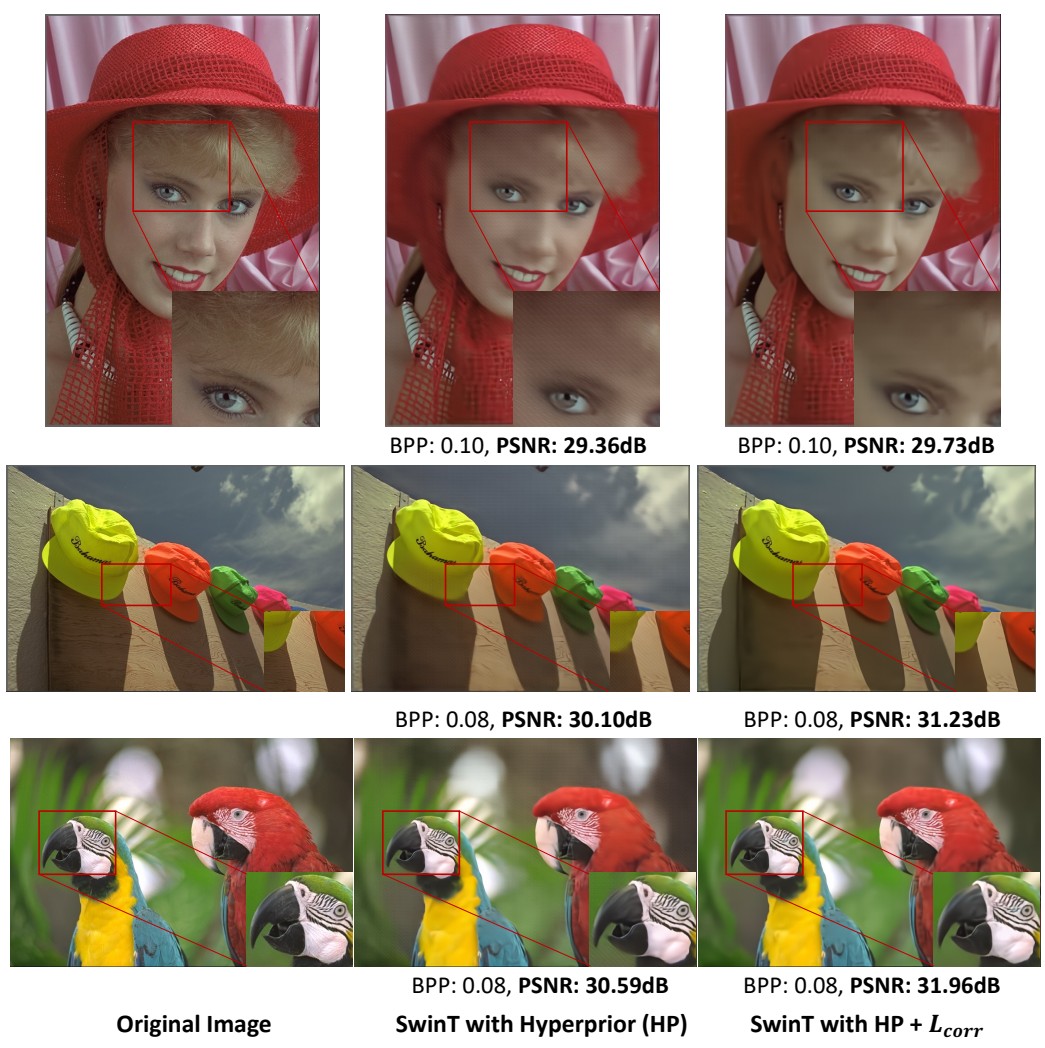

BPP: 0.10, **PSNR: 29.36dB**     BPP: 0.10, **PSNR: 29.73dB**

BPP: 0.08, **PSNR: 30.10dB**     BPP: 0.08, **PSNR: 31.23dB**

BPP: 0.08, **PSNR: 30.59dB**     BPP: 0.08, **PSNR: 31.96dB**

**Original Image**     **SwinT with Hyperprior (HP)**     **SwinT with HP + $L_{corr}$**

Figure 8: Comparison of SwinT Hyperprior and SwinT Hyperprior with correlation loss on reconstructed images kodim04, kodim03, and kodim23 from the Kodak dataset.

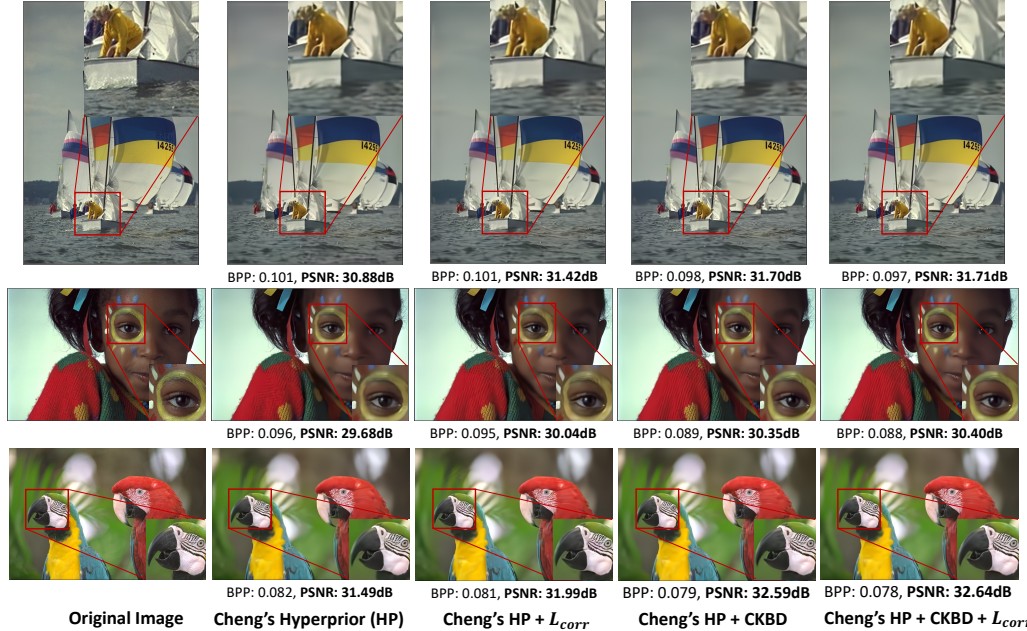

BPP: 0.101, **PSNR: 30.88dB**   BPP: 0.101, **PSNR: 31.42dB**   BPP: 0.098, **PSNR: 31.70dB**   BPP: 0.097, **PSNR: 31.71dB**

BPP: 0.096, **PSNR: 29.68dB**   BPP: 0.095, **PSNR: 30.04dB**   BPP: 0.089, **PSNR: 30.35dB**   BPP: 0.088, **PSNR: 30.40dB**

BPP: 0.082, **PSNR: 31.49dB**   BPP: 0.081, **PSNR: 31.99dB**   BPP: 0.079, **PSNR: 32.59dB**   BPP: 0.078, **PSNR: 32.64dB**

**Original Image**   **Cheng's Hyperprior (HP)**   **Cheng's HP + $L_{corr}$**   **Cheng's HP + CKBD**   **Cheng's HP + CKBD + $L_{corr}$**

Figure 9: Comparison of Cheng's Hyperprior (HP), Cheng's HP with correlation loss, Cheng's HP with Checkerboard (CKBD), and Cheng's HP with CKBD and correlation on reconstructed images kodim09, kodim15, and kodim23 from the Kodak dataset.