# OpenReview forum: "Towards Efficient Image Compression Without Autoregressive Models"
_NeurIPS.cc/2023/Conference — NeurIPS 2023 poster_

### Official Review · Reviewer_qpdX · 2023-06-30

**Soundness:** 2 fair
**Presentation:** 2 fair
**Contribution:** 2 fair
**Rating:** 5
**Confidence:** 4

**Summary:**

This paper aims at providing a efficient and effective entropy model to achieve better trade-off between performance and complexity for learned image compression. They introduce a correlation loss to force the latent to be spatially decorrelated so that it can fit the independent probability model better.

**Strengths:**

1. While most existing methods primarily focus on the context model, this paper makes a try at decorrelating latents with the proposed correlation loss. The correlation loss can act as a general plug-in for the hyperprior-based methods, which is flexible.

2. It is proved that when the proposed method is combined with the checkerboard method,  the performance gain is about 85% compared with the auto-regressive model with only 1/50th inference time, which is efficient and effective.

**Weaknesses:**

1.  As shown in the RD curves, the BD-rate gains are obvious in the low bit-rate regime, however, there are less or even no gains in the high bit-rate regime. I think these results require further analysis, such as why it works better in the low bit-rate regime. It is one of the core problems that should be answered to help readers better understand the method.

2.  It is just a little problem. There are many references of the figures and equations in the paper, but they are somewhat orderless. It is better to put the figures not so far away from their references.

3.  I am somewhat confused about the analysis about Figure 4. It is better to provide the input image, which can help readers know the correspondence between the visualization and the original input.

4.  I am not sure about the claim in line 307 "The fourth column of Figure 4 shows that for our method, latent space has significantly reduced correlation compared to the baseline indicating the correlation loss’s efficacy". I can only see that compared with Cheng's Hyperprior, the normalized latent of the proposed method has lower energy. I do not think it can provide evidence to show the proposed approach will reduce the correlation. Maybe it is because the proposed method just gives more accurate mean and variance. I think you can also provide the visualization of using auto-regressive context model but without the proposed approach. It may provide similar visualization results.

**Questions:**

No

**Limitations:**

see the weakness part

---

> ### Author Rebuttal · Authors · 2023-08-09
>
> We thank the reviewer for their comments and suggestions.
>
> > As shown in the RD curves, the BD-rate gains are obvious in the low bit-rate regime, however, there are less or even no gains in the high bit-rate regime. I think these results require further analysis, such as why it works better in the low bit-rate regime. It is one of the core problems that should be answered to help readers better understand the method.
>
> Please refer to the general response for a detailed answer to this question
>
> > It is just a little problem. There are many references of the figures and equations in the paper, but they are somewhat orderless. It is better to put the figures not so far away from their references.
>
> We appreciate the valuable feedback provided by the reviewer. We will address and correct these errors in the camera-ready version of the paper. Thank you for bringing these to our attention.
>
> > I am somewhat confused about the analysis about Figure 4. It is better to provide the input image, which can help readers know the correspondence between the visualization and the original input.
>
> Please refer to the Figure 1 in the attached PDF, where we have included the input image for a better understanding of the correspondence between the visualization and the original input. We shall update all the Figures, including Figure 4 in the main manuscript and Figure 5 in the Supplementary, in the camera-ready version of our paper to ensure clarity and accuracy. Your feedback is greatly appreciated.
>
> > I am not sure about the claim in line 307 "The fourth column of Figure 4 shows that for our method, latent space has significantly reduced correlation compared to the baseline indicating the correlation loss’s efficacy". I can only see that compared with Cheng's Hyperprior, the normalized latent of the proposed method has lower energy. I do not think it can provide evidence to show the proposed approach will reduce the correlation. Maybe it is because the proposed method just gives more accurate mean and variance. I think you can also provide the visualization of using auto-regressive context model but without the proposed approach. It may provide similar visualization results.
>
> We appreciate the reviewer's feedback regarding the interpretation of Figure 4 and the need for improved clarity, particularly concerning the representation of reduced correlation of the latent variable y in the last column of our figures. We have given thoughtful attention to this matter and have taken measures to enhance the clarity of the Figure, as demonstrated in the Figure 1 of the attached PDF. We aim to offer a more straightforward and comprehensive visualization of the effects of correlation loss. For a concise and coherent explanation, we kindly direct the reviewer to the general response section.

---

### Official Review · Reviewer_vwAa · 2023-07-04

**Soundness:** 3 good
**Presentation:** 2 fair
**Contribution:** 3 good
**Rating:** 5
**Confidence:** 5

**Summary:**

This paper proposed a correlation loss to decrease the correlation among spatially-neighbored elements in the latent space. By only modifying the loss function, this method acts as a plug-in method for the existing neural compression methods with no complexity increasing. Experiments show improvement in the compression performance to several baseline models.

**Strengths:**

The main contribution of the paper is the correlation loss proposed to decrease the correlation among spatially in latent features.

Experiments show improvement in the compression performance to several baseline models which provide some insights about designing better neural image compression networks from the prospective of feature map decorrelation.

**Weaknesses:**

1) For SwinT and Cheng with checkerboard, the correlation loss seems to only work for lower bitrates, the author may provide some explanation. Besides, The current tested bitrate range is relatively low. RD performance at higher bitrates (>1bpp) should also be provided.

2) The paper writing can be improved. The citation format in the article looks messy. For equations, Some are ‘eq’, but some are ‘Equation.’

3) Experiments on factorized Ball\'e method [1] and HP+AR+correlation loss can be provided to makes the experiments and evaluation more complete.

[1] Johannes Ballé, Valero Laparra, and Eero P Simoncelli. End-to-end optimized image compression.

**Questions:**

Minor issues:

1) Some experiments in the supplementary materials are kind of important and can be put into the main paper.

2) Some citations are still preprint version (e.g. [1][2][3] in reference). Their officially published version should be cited.

---

> ### Author Rebuttal · Authors · 2023-08-09
>
> We thank the reviewer for their comments and suggestions.
>
> > For SwinT and Cheng with checkerboard, the correlation loss seems to only work for lower bitrates, the author may provide some explanation
>
> Please refer to our general response for a detailed explanation.
>
> > Besides, The current tested bitrate range is relatively low. RD performance at higher bitrates (>1bpp) should also be provided.
>
> Our experimentation focused on a bits per pixel (bpp) range spanning from 0.05 to 0.6, corresponding approximately to a peak signal-to-noise ratio (PSNR) range of 25 to 35. In alignment with this approach, Cheng's and Checkerboard methods also adhere to a comparable range, operating within 0.1 to 0.8 bpp, which roughly corresponds to a PSNR range of 27 to 37.
>  We also observed from the recent works that as the compression rate increases, lossy compression approaches the realm of lossless compression, where there is limited potential for coding gain through improved predictions, as the codec must encode the inherent noise (unpredictable) within the data [5]. This notion aligns with the modest enhancements observed in generative modeling, such as the slight improvement in negative log-likelihood as depicted in Figure 1 (a) of [3], or the minor bit-per-dim discrepancy in recent learned lossless image compression, as shown in Table 1 of [4].
> To align with the trends observed in recent research, we concentrated our experiments within widely adopted ranges where significant improvements have been reported.
> However, we are committed to incorporating more bpp points for our experiments, particularly for SwinT and Minnen's hyperprior-based methods, in the camera-ready version of the paper. This will allow us to provide a comprehensive evaluation of these methods across a broader range of bpp values.
>
>
> [1] Cheng, Z., Sun, H., Takeuchi, M., & Katto, J. (2020). Learned image compression with discretized gaussian mixture likelihoods and attention modules. In Proceedings of the IEEE/CVF conference on computer vision and pattern recognition (pp. 7939-7948).
>
> [2] He, D., Zheng, Y., Sun, B., Wang, Y., & Qin, H. (2021). Checkerboard context model for efficient learned image compression. In Proceedings of the IEEE/CVF Conference on Computer Vision and Pattern Recognition (pp. 14771-14780).
>
> [3] Kingma, D., Salimans, T., Poole, B., & Ho, J. (2021). Variational diffusion models. Advances in neural information processing systems, 34, 21696-21707.
>
> [4] Berg, R. V. D., Gritsenko, A. A., Dehghani, M., Sønderby, C. K., & Salimans, T. (2020). Idf++: Analyzing and improving integer discrete flows for lossless compression. arXiv preprint arXiv:2006.12459.
>
> [5] Zhu, Y., Yang, Y., & Cohen, T. (2021, October). Transformer-based transform coding. In International Conference on Learning Representations.
>
> > Experiments on factorized Ball'e method [1] and HP+AR+correlation loss can be provided to makes the experiments and evaluation more complete.
>
> As recommended by the reviewer, we will incorporate Balle's Factorized prior [1] and Balle's Hyperprior (HP) [2], along with HP + Correlation Loss and HP + AR, in the camera-ready version of our paper. This addition will provide a comprehensive analysis and further insights into our proposed approach.
>
> [1] Ballé, Johannes, Valero Laparra, and Eero P. Simoncelli. "End-to-end optimized image compression." arXiv preprint arXiv:1611.01704 (2016).
>
> [2] Ballé, Johannes, David Minnen, Saurabh Singh, Sung Jin Hwang, and Nick Johnston. "Variational image compression with a scale hyperprior." arXiv preprint arXiv:1802.01436 (2018).
>
> > Some experiments in the supplementary materials are kind of important and can be put into the main paper.
>
> > Some citations are still preprint version (e.g. [1][2][3] in reference). Their officially published version should be cited.
>
> > The paper writing can be improved. The citation format in the article looks messy. For equations, Some are ‘eq’, but some are ‘Equation.’
>
> We thank the reviewer for the valuable comments, we shall fix these errors in the camera-ready version of the paper.

---

> > ### Comment · Reviewer_vwAa · 2023-08-18
> >
> > Thank you for your response. I will stick to my rating.

---

### Official Review · Reviewer_7bvo · 2023-07-07

**Soundness:** 4 excellent
**Presentation:** 3 good
**Contribution:** 3 good
**Rating:** 6
**Confidence:** 4

**Summary:**

This paper presents a new loss to improve neural image compression models. The authors identify a potential issue with typical neural image compression models where the hyperprior, which predicts entropy parameters over a quantized latent representation, assumes conditional independence between latents, but that may not be true in practice. Previous methods improved the entropy model through context modeling (e.g., using a spatially autoregressive model or a "checkerboard" decomposition) but this leads to higher compute and slower runtimes.

Instead, the authors add a local correlation loss averaged spatially over the latents. This loss directly encourages the encoder to generate decorrelated latents, which is a better match for the conditional independence assumption built in to the hyperprior. In particular, the loss affects training (and isn't very expensive) and adds no additional computation at inference time, which leads to significant runtime improvements.

The paper shows that the new correlation loss improves rate-distortion (RD) performance when applied to several popular models (see Fig. 1 and Fig. 7).

**Strengths:**

The primary strength of the paper is the simplicity and effectiveness of the main idea.

Most papers on neural compression boost RD performance by making the model more complex, which typically leads to slower decode times. The authors discuss this problem and present a well-motivated loss that leads to significant RD gains without affecting runtime. Their correlation loss is also quite general and can be applied to many different compression models.

As far as I know, the correlation loss has not been proposed elsewhere in the neural compression literature. There are some similarities to a diversity loss in VQ or clustering, though I don't have a specific reference for this. So I think the originality, especially for the neural compression subfield, is high.

The quality and clarity of the writing is also high.



**Weaknesses:**

Ideally, the correlation loss presented in the paper would be applied to a SOTA comperssion model leading to a new SOTA. For instance, the paper cites (He 2021), which introduced the checkerboard decomposition for entropy modeling, but they don't build on top of (He 2022), which presents a more powerful model that combines the checkerboard with CHARM.

ELIC: Efficient Learned Image Compression with Unevenly Grouped Space-Channel Contextual Adaptive Coding
Dailan He, Ziming Yang, Weikun Peng, Rui Ma, Hongwei Qin, Yan Wang
https://arxiv.org/abs/2203.10886

Figure 1 implies that the benefit of the correlation loss shrinks as models become more powerful so it may be that the benefit for ELIC is negligible?

The visualization in Fig. 4 is great (and is common for compression papers) but it's not obvious to me what I should be looking at to see the impact of the correlation loss. Is it a sharper scale image? The goal is an i.i.d. Gaussian normalized image (far right column) but it's not visually obvious to me that "our approach" is closer to i.i.d. Gaussian than the baseline.

Mask patterns are mentioned and shown in Fig. 6 but results are only in the supplemental material. Presumably that's because the mask shape did not have a large impact. That's fine, but maybe cut Fig. 6 or at least add a sentence to the main paper summarizing the findings.



**Questions:**

Analysis exploring why the correlation loss boosts performance would strengthen the paper. Specifically, why is it needed when the existing rate-distortion loss is minimized by decorrelated latents? The generic answer is "the network is stuck in a (bad) local minimum, and the correlation loss changes the loss landscape such that the optimizer doesn't get stuck".

**Limitations:**

adequately addressed

---

> ### Author Rebuttal · Authors · 2023-08-09
>
> We thank the reviewer for their comments and suggestions.
>
>
> > Ideally, the correlation loss presented in the paper would be applied to a SOTA compression model leading to a new SOTA. For instance, the paper cites (He 2021), which introduced the checkerboard decomposition for entropy modeling, but they don't build on top of (He 2022), which presents a more powerful model that combines the checkerboard with CHARM.
> ELIC: Efficient Learned Image Compression with Unevenly Grouped Space-Channel Contextual Adaptive Coding Dailan He, Ziming Yang, Weikun Peng, Rui Ma, Hongwei Qin, Yan Wang https://arxiv.org/abs/2203.10886
>
> Please refer to the “Performance on ELIC: Efficient Learned Image Compression” in General Response for Common Comments Section
>
> > Figure 1 implies that the benefit of the correlation loss shrinks as models become more powerful so it may be that the benefit for ELIC is negligible?
>
> The primary advantage of the correlation loss lies in its capacity to minimize the discrepancy between the actual and presumed probability distribution within the entropy model. Figure 2 in the attached PDF illustrates that for a given encoder and decoder architecture, which is the Cheng's Hyperprior (CH) in this case, the choice of entropy model determines performance of the resulting model. Note that CH signifies the lower bound with the given encoder-decoder architecture, whereas Cheng’s AR defines the upper limit with the same encoder-decoder architecture. When correlation loss is applied to CH, significant improvements can be achieved due to the large performance gap between the CH baseline and the upper limit. On the other hand, since the gaps are diminished in the models like CH + CKBD and CH + ChARM, the pothential for improvement by applying correlation loss are also limited when compared to the CH baseline. In summary, the introduction of correlation loss to these models would result in a closer approximation to the full AR performance, albeit with gains that are comparatively smaller.
>
> We expect the similar pattern might be observed if we change the encoder-decoder architecture from CH's to ELIC's. Note that ELIC not only impove the entropy model by the proposed space-channel context model (SCCTX), which combines CKBD and ChARM, but also introduced architectural modification of the encoder and decoder, which contribute an additional BD rate gain of approximately 8-12% [1]. While we expect that the performance of Cheng's with SCCTX might be lower than ChARM with correlation loss, the inclusion of correlation loss has the potential to elevate it to a level comparable to, or akin to, the full AR performance, all while costing only about 1/15th of the computation expense (2x of ChARM).
>
> [1] He, D., Yang, Z., Peng, W., Ma, R., Qin, H., & Wang, Y. (2022). Elic: Efficient learned image compression with unevenly grouped space-channel contextual adaptive coding. In Proceedings of the IEEE/CVF Conference on Computer Vision and Pattern Recognition (pp. 5718-5727).
>
> > The visualization in Fig. 4 is great (and is common for compression papers) but it's not obvious to me what I should be looking at to see the impact of the correlation loss.
>
> We acknowledge the reviewer's observation that the interpretation of Figure 4 lacks clarity, particularly in relation to the depiction of the reduced correlation of the latent variable y in the last column of our Figures (main manuscript [Figure 4], supplementary material [Figure 5]). We have given careful consideration to this concern and have taken steps to enhance the clarity of the Figure (please see Figure 1 in the attached PDF). Our goal is to provide a more straightforward and comprehensive illustration of the impact of correlation loss. For a more concise and coherent explanation, we kindly refer the reviewer to the general response section.
>
> > Mask patterns are mentioned and shown in Fig. 6 but results are only in the supplemental material.
>
> We highly value the constructive feedback provided by the reviewer and are committed to incorporating these suggested changes into the camera-ready version of our paper.
>
> > Analysis exploring why the correlation loss boosts performance would strengthen the paper.
> >  Specifically, why is it needed when the existing rate-distortion loss is minimized by decorrelated latents? The generic answer is "the network is stuck in a (bad) local minimum, and the correlation loss changes the loss landscape such that the optimizer doesn't get stuck".
>
> The primary focus of our correlation loss is directed at diminishing the correlation present among neighboring elements within the latent space, as illustrated in Figure 1 of the attached PDF. This reduction in correlation assumes a critical role in mitigating the disparities between the assumed probability distribution of the hyperprior entropy model and the actual distribution of latent variables. The effect of decreased correlation is illustrated in the figure and the detailed analysis can be found in the text of the our main rebuttal section.
>
> We agree there is high possibility that the correlation loss introduces alterations to the loss landscape. However, it remains uncertain due to the current lack of concrete theoretical evidence about exact interplay between the correlation loss, rate loss, and the distortion loss. Thus, in the current circumstances, we cannot definitively assert whether this is indeed the case.

---

### Official Review · Reviewer_xepj · 2023-07-12

**Soundness:** 4 excellent
**Presentation:** 4 excellent
**Contribution:** 4 excellent
**Rating:** 8
**Confidence:** 5

**Summary:**

This paper focus on efficient learned image compression. Different from existing method, which generally aims to parallelize the autoregressive operations, this paper propose to speed up the framework by removing the whole autoregressive model by introducing the correlation loss, aiming to decorrelate the latent features. The introduced correlation loss can act as a plug-in for the existing learned image compression methods to achieve superior RD performance and at the same time reduce the inference time without autoregressive entropy model.

**Strengths:**

1. The paper is well-motivated and easy to follow.
2. The introduced correction loss is statistically sound and empirically proved to be effective. The proposed loss is novel to me and can act a plug-in for the existing learned image compression to achieve consistent improvement, especially at low-bit ranges.
3. By removing the correlation, the propose method can ease the requirement of autoregressive entropy model, so as to speed up the whole compression framework.

**Weaknesses:**

1. The citation in Figure 3 is wrong.
2. The caption of Figure 4 states that the correlation loss can provide more flexible parameterized distribution models with significant spatial redundancy reduction. However, from my perspective, the plots show in Figure 4 cannot showcase both "more flexible parameterized distribution models" and "significant" spatial redundancy reduction. I hope the author can have more comments on this, otherwise such claims are unsupported or inaccurate.
3. The introduction of section 3.1 should be reduced, which are general, well-known concepts in LIC, the main focus with more words should be put into the contribution introduced section 3.2.
4. The paper should also report the performance of the proposed method on top of some more recent efficient LIC methods like [1].

[1] Dailan He, Ziming Yang, Weikun Peng, Rui Ma, Hongwei Qin, and Yan Wang. ELIC: Efficient Learned Image Compression with Unevenly Grouped Space-Channel Contextual Adaptive Coding. CVPR 2022

**Questions:**

1. Is the hyperprior shown in Figure 4 a GMM-based (K=3) or a simple mean-scale hyperprior? If it is GMM, the plots shown in Figure 4 is only one latent feature of the K=3? I think this should be specified to be clearer to readers.
2. Will the code be released to the general public?

---

> ### Author Rebuttal · Authors · 2023-08-09
>
> We thank the reviewer for their comments and suggestions.
>
> > The caption of Figure 4 states that the correlation loss can provide more flexible parameterized distribution models with significant spatial redundancy reduction. However, from my perspective, the plots show in Figure 4 cannot showcase both "more flexible parameterized distribution models" and "significant" spatial redundancy reduction. I hope the author can have more comments on this, otherwise such claims are unsupported or inaccurate.
>
> We acknowledge the reviewer's observation regarding the clarity of the caption of Figure 4 and the accompanying text, and we appreciate the feedback. To rectify this concern and provide a clearer presentation of our intended message, we have undertaken significant revisions to the Figure itself, which can be referred to in the attached PDF document (Figure 1). Moreover, we have taken comprehensive steps in our general response to thoroughly address all the concerns raised by the reviewers concerning Figure 4.
>
> > The paper should also report the performance of the proposed method on top of some more recent efficient LIC methods like [1].
>
> Please refer to the “Performance on ELIC: Efficient Learned Image Compression” in General Response for Common Comments Section
>
> > Is the hyperprior shown in Figure 4 a GMM-based (K=3) or a simple mean-scale hyperprior? If it is GMM, the plots shown in Figure 4 is only one latent feature of the K=3? I think this should be specified to be clearer to readers.
>
>  The hyperprior shown in Figure 4 is a simple mean-scale hyperprior.
>
> > Will the code be released to the general public?
>
> We shall release the code with the camera-ready version of the paper.
>
> > The citation in Figure 3 is wrong.
>
> >The introduction of section 3.1 should be reduced, which are general, well-known concepts in LIC, the main focus with more words should be put into the contribution introduced section 3.2.
>
> We express our gratitude to the reviewer for their valuable insights, comments, and suggestions, which have greatly contributed to the improvement of our paper. We are committed to incorporating these recommended changes and enhancements in the final version of the paper to ensure its quality and accuracy.

---

> > ### Comment · Reviewer_xepj · 2023-08-18
> >
> > I appreciate the responses from the authors, which address my concerns and the clarity issue. I decide to raise my rating.

---

### Author Rebuttal · Authors · 2023-08-09

We thank all the reviewers for their insightful comments and suggestions.

# Additional Experiment Results

During the rebuttal period, we have performed additional experiments as shown in Figure 2 of the attached PDF, which will be the updated version of the Figure 1 of the main manuscript. The updated results include enhancements to the performance of the Checkerboard (CKBD) model with the integration of correlation loss, as well as the application of correlation loss to the ChARM model.
With the incorporation of correlation loss, the CKBD model now exhibits a BD rate gain of 16.5% when compared to Cheng’s Hyperprior (CH), representing around 90% of the performance achieved by a full AR model, all at approximately 1/50th of the computational cost.
Furthermore, the utilization of correlation loss with the ChARM model results in a significant BD rate gain of 18% over the baseline Cheng’s Hyperprior (CH). This gain corresponds to approximately 98% of the improvement obtained through the use of a full AR model. Notably, this progress is achieved with a considerably reduced computational cost, amounting to roughly 1/30th of the full AR cost.
Figure 2 in the attached PDF also serves to visually underline the implications of these findings. It demonstrates that a fixed encoder and decoder architecture lays down performance boundaries defined by the chosen entropy model. At one end, Cheng's Hyperprior (CH) signifies the lower performance limit, while at the other, Cheng's AR defines the upper threshold for given the CH's encoder-decoder architecture. With correlation loss incorporated into CH, significant improvements are attained. Note that CH + CKBD and CH + ChARM offer performance gains over CH closer to the upper limit. Consequently, the room for enhancement by adding correlation loss in these cases is relatively constrained when compared to the CH model. Nonetheless, applying correlation loss to these models leads to better approximations of full AR performance.

# General Response for Common Comments:

## Analysis on less gains in the high bit-rate regime is required. {Reviewers: 7bvo, qpdX}

One limitation of our proposed method, as mentioned in Section 5 of the main manuscript, is that the efficacy of the proposed loss function tends to be less pronounced at higher bit rates, despite the considerable performance gains at lower bitrates. This behavior can be attributed to the characteristics of learned image compression (LIC) models, which exhibit higher correlation among latent variables at lower bit-per-pixel (bpp) values compared to higher bpp values, as reported by Zhu et al. [1] and also evident from the correlation maps in Figure 4 of the supplementary material.
We conducted a comprehensive analysis and presented the findings in Figure 3 of the attached PDF. This Figure showcases the relationship between PSNR gains, bpp, and correlation for different models, including Cheng's Hyperprior (CH), CH + CKBD, SwinT Hyperprior, and Minnen's Hyperprior. From the graphs in the Figure 3, a clear pattern can be observed: as the bpp decreases, the correlation of the latents increases, resulting in a higher gain in PSNR. However, as we move towards higher bpps, the correlation becomes notably reduced, resulting in decreased PSNR gain. This trend explains the reason why the efficacy of the correlation loss is larger in low bpp range and diminish at high bpp range.

[1] Transformer-based transform coding, ICLR. 2021.


## The visualization of Figure 4 does not explain the impact of Correlation loss. {Reviewers: xepj,7bvo, qpdX}

The reviewers have pointed out that the last column in all our Figures (Figure 4 in the main manuscript and Figure 5 in the supplementary material) may not clearly illustrate the reduced correlation of the latent variable y. While we acknowledge this limitation, the bpp values and correlation maps in Figure 4 (Supplementary material) provide compelling evidence of the reduced correlation in the latent space achieved through the application of correlation loss. To address this concern and provide a better visual demonstration, we have included the correlation map and the corresponding total correlation value of the latent space in the Figure 1 of attached PDF.
The Figure 1 illustrates that the improved mean μ and scale σ effectively capture image structures with notable superiority, incurring a minimal extra cost of 0.001 bpp. Similarly, the latent variable y undergoes a reduction of around 0.002 bpp, stemming from a correlation decrease of roughly 3.5 times. These combined enhancements manifest in a PSNR gain of approximately 0.38 dB when compared to the baseline, all while ensuring a total bpp that remains lower than the baseline.

## Performance on ELIC: Efficient Learned Image Compression: {Reviewer: xepj, 7bvo}

We conducted experiments on ELIC (using the unofficial implementation available on Github), investigating its performance both with and without the integration of the correlation loss. A comparison between ELIC and Cheng's Hyperprior revealed a significant BD rate gain of 28.85% for ELIC. Intriguingly, when the correlation loss was introduced to ELIC, the BD rate gain was further elevated to approximately 30.36%.
It is worth highlighting that the complete inference process for the ELIC on Kodak dataset took approximately 17.77 seconds which is about 1/15th of the total inference time of full AR. These findings strongly underscore the effectiveness of our proposed approach, showcasing its potential to achieve enhanced image compression performance.


### If given the opportunity to revise our original manuscript, we will update the Figures and text to clearly convey our new results and findings in this rebuttal and improve the overall clarity of our message. We appreciate the reviewers' feedback and are committed to presenting our research in the best possible way in the final version of the paper.

---

### Decision · Program_Chairs · 2023-09-21

**Decision:**

Accept (poster)

**Comment:**

The paper proposes a new loss to improve neural image compression models. The paper shows that the loss improves rate-distortion performance for several popular models.

After the rebuttal, all reviewers recommend to accept the paper. The reviewers agree that the paper provides a simple and effective idea for improving image compression models. The reviewers also agree that the idea is sufficiently validated and shown to be effective for improving image compression models.

The reviewers also identified minor issues that were for the most part addressed by the rebuttals of the authors.